# A Framework for the Development of Wetland for Agricultural Use in Indonesia

**Andi Amran Sulaiman [1], Yiyi Sulaeman [2],\* and Budiman Minasny [3]**

[1]   Faculty of Agriculture, Hasanuddin University, Makassar, Sulawesi Selatan 90245, Indonesia; adcmentan73@gmail.com
[2]   Indonesian Center for Agricultural Land Resource Research and Development, Bogor, Jawa Barat 16114, Indonesia
[3]   School of Life and Environmental Sciences, The University of Sydney, Eveleigh, NSW 2015, Australia; budiman.minasny@sydney.edu.au
\*   Correspondence: yiyisulaeman@pertanian.go.id; Tel.: +62-251-832-3012

**Abstract:** Crop production needs to double to feed the world's growing population. Indonesia, as the fourth most populated country in the world, needs to meet its food security challenge with a shrinking arable land area. Indonesia has over 34 million ha of swampland. The scarcity of arable land in Indonesia means wetlands are likely to be converted to agricultural use. The challenge is to both profitably and sustainably do so. This paper presents a framework for developing wetlands for food production, which includes (1) the characterization of land and problem of development; (2) analysis of historical development and lessons learned; (3) technology development; and (4) optimization of development. We analyze each of the components and its relation to regional economic growth and lessons learned. For successful future wetland development, three factors must be considered: Land-soil-water characterization, landscape and land use design, and community development. This framework can be adopted by other tropical areas for the development of wetlands.

**Keywords:** wetland; food security; peatland; swamp

## 1. Introduction

The world's population is growing, and by 2050, it is estimated to increase by more than 35%. To feed this growing population, crop production will need to double [1]. In Indonesia, competing interests are affecting food security [2,3]. Indonesia has 264 million people [2] and an annual population growth rate of 1.3%, but shrinking land area per capita and levelling crop productivity. As the population grows, so does rice consumption, however, agricultural land is fragmented. Some of the most productive agricultural lands have been converted to residential, commercial, or other purposes. In the face of climate change and extreme weather conditions, Indonesia needs to find ways to increase crop production [4].

The Indonesian government has set a target of food self-sufficiency and becoming the world's food basket by 2045 [5]. As arable land is limited, one option to increase food production is to convert wetlands [6]. As an archipelago, Indonesia has 99,093 km of shoreline, the second longest in the world after Canada [7]. If managed properly, the coastal zone can provide an extensive agricultural area. There are many types of wetlands, and in Indonesia, developing swampland is the focus. The development of wetlands creates favorable conditions for social and economic development such as agriculture, fisheries, aquaculture, etc. An example is in Vietnam, in which the development of agriculture in the Mekong Delta has improved livelihood and reduced poverty [8]. In this paper, we

used the term reclaim with the definition: "to make available for human use by changing natural conditions" [9].

While wetlands have been reclaimed for agriculture in many parts of the world, they are not an ideal land resource for agriculture. They have:

- important environmental and ecosystem services [10,11],
- high pests and diseases threat [12],
- a high water table for extended periods (prone to flooding) [13],
- saltwater intrusion (for coastal tidal swamps) [12,14],
- a low pH of 3 to 4.5 [12],
- pyrite-affected acid sulfate soils [15],
- high concentrations of toxic elements such as iron (Fe), aluminium (Al), sulphur (S), and sodium (Na) [15], and
- deficiencies in nutrients such as phosphorus (P), potassium (K), zinc (Zn), copper (Cu), and boron (B) [12].

However, there are also advantages in developing swamplands in Indonesia. They are relatively available, have abundant water, are resilient to seasonal drought, and provide a longer cropping period [12]. Swamplands have great potential to be used as an integrated farming system (food crops, estate crops, and animal husbandry). The scarcity of arable land in Indonesia means that wetlands are likely to be converted to agricultural use. The challenge is to do so both profitably and sustainably [6].

Agro-ecosystems in Indonesia are divided into rice fields (irrigated and rainfed), dryland (dry climate and wet climate), and wetland (tidal and non-tidal swamp agricultural land). Reclamation is the first step in developing swamps for agriculture. This reclamation process plans for water management to accelerate the ripeness of soil, so that crops and land management can be established.

For hundreds of years, the indigenous Banjar people in Kalimantan have settled in swamps. They built small ditches that protrude 2–3 km out from a stretch of river. The Dayak people of South Kalimantan also used peatlands in the swamp for growing rice, which they called lawau [16]. These indigenous community practices have succeeded in developing swamp areas into rice, secondary crops, vegetables, and other horticulture crops in Kalimantan and Sumatra. The success of the community use of swampland inspired the government to reclaim more swampland for agriculture.

A study on wetland loss in Vietnam reminded us to learn from past successes and failures so that future management should take a holistic approach that includes a better understanding of the implications of past decisions [17].

This paper aims to present the framework of swamp development in Indonesia for agriculture. In order to understand the complexity of swamp development, we need to understand its history of development, which can be divided into five periods; before independence (pre-1950), 1956–1958, 1969–1995, 1995–1999, and 2000–2013. Finally, we discuss a general framework and suggest some ways forward to use swamplands sustainably.

## 2. Framework Development

The conceptual framework for the development of swampland in Indonesia is presented in Figure 1. The framework is composed of four stages, which can be considered as steps to be followed for its application.

1. Characterization of land and problem of development
2. Analysis of historical development and lessons learned
3. Technology development
4. Optimization of development

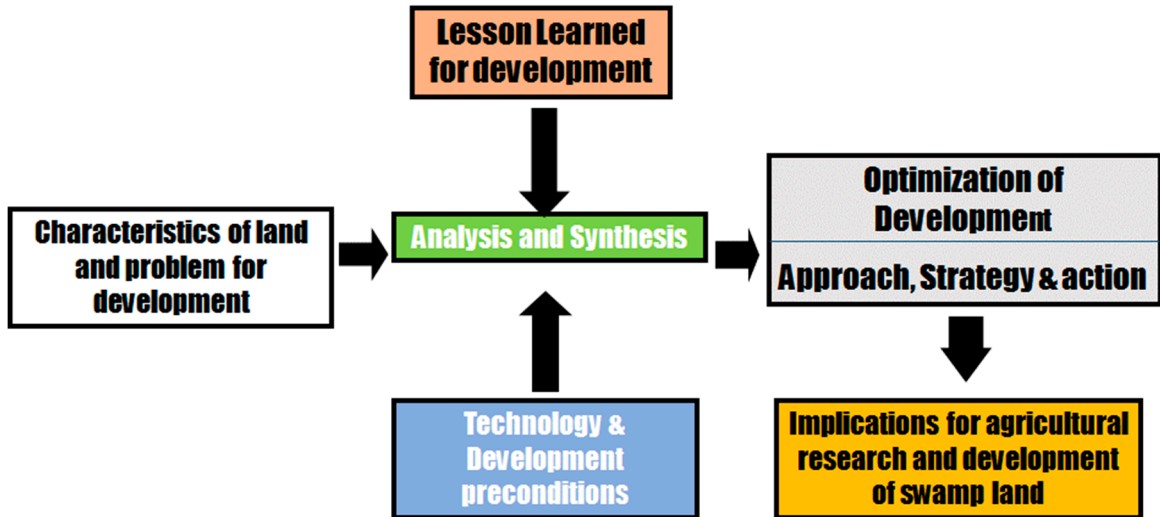

**Figure 1.** A conceptual framework for swampland development in Indonesia, based on Reference [18].

In the following sections, we will discuss each of the components and then discuss aspects that allow for the implementation of the framework.

## 3. Characterization of Swamplands in Indonesia

The distribution of swampland in Indonesia is illustrated in Figure 2. Swampland (rawa) in Indonesia is mainly divided into two types: Tidal swampland (rawa pasang surut) and non-tidal swamp or inland swampland (rawa lebak). Peats can be found in both types of swampland. However, for mapping purposes, peatland (lahan gambut) is separated from swampland.

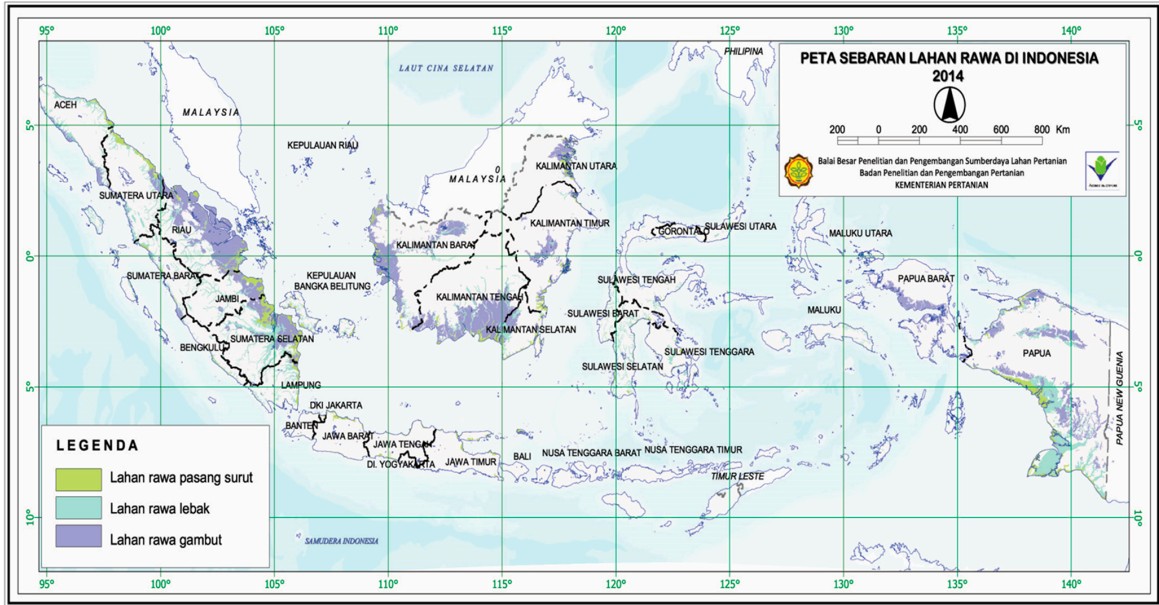

**Figure 2.** Distribution of swampland in Indonesia (source: Indonesian Center for Agricultural Research and Development). Lahan rawa pasang surut = tidal swamp, lahan rawa lebak = non-tidal swamp, and lahan rawa gambut = peatland.

Tidal swamps are part of the coastal plain influenced by tidal fluctuations. The water level in tidal swampland rises during the rainy season. In Kalimantan, this usually starts in October and peaks in January or February. The water level drops in March or April and remains stagnant until June.

The water table is lowest during the dry season, from June to October. Tidal swamps can be divided into four types based on the degree of tidal influence [19]:

Class A. Areas that are directly affected by sea tides and can be flooded at high tide for at least 4–5 days during a 14-day spring tide cycle. This class supports double cropping of wetland rice if salinity intrusion is less than two months/year.

Class B. Areas that are influenced by sea tides but only flooded during spring. This class supports a single tidal irrigated wetland rice crop, possibly followed by a legume.

Class C. Areas not directly affected by sea tides and not flooded during high tide. The area is affected by high groundwater levels.

Class D. Higher elevation areas unaffected by sea tides.

Non-tidal swamps (lebak) are river floodplains that are not influenced by sea tides. Crops are usually planted at the end of rainy season. Non-tidal swamplands can be divided into three categories based on their topography and the duration of water inundation:

- Shallow swamp or embankment (Lebak pematang): Land on a natural levee with relatively high topography and a shallow or short inundation.
- Deep swamp (Lebak dalam): Land located far off the coast, in a basin, that is continuously and deeply inundated.
- Mid swamp (Lebak tengah): Land that is located between the inland and the ridge of the embankment.

Figure 3 illustrates a cross-section of a non-tidal swamp area.

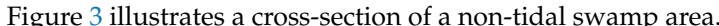

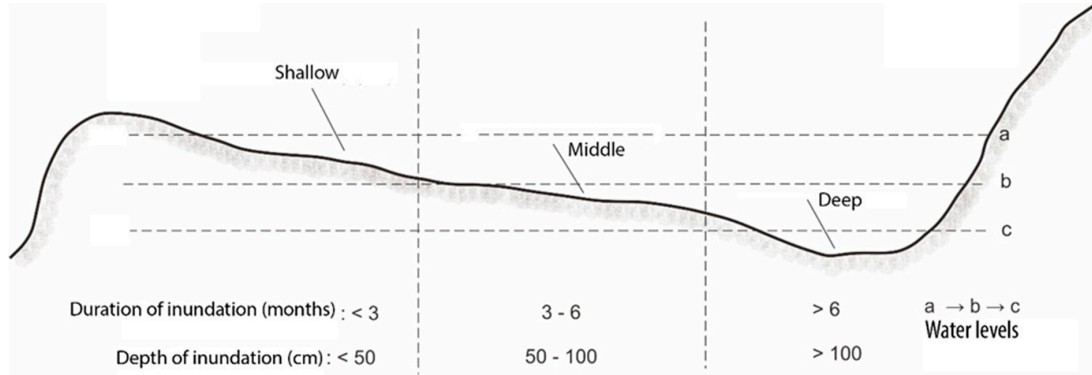

**Figure 3.** A cross-section of a non-tidal swamp area.

Soils in swamplands are marine or fluvial deposits, and peats. Soils that have developed on marine sediments include actual and potential acid sulphate soils. Some of the peats were deposited above pyritic sediments. The pyrite layer is naturally present in marine sediments. Under waterlogged conditions, it is in a reduced condition and if the pyrite layer is exposed to air and oxidized, the pH of the soil becomes extremely acidic [19].

There are about 34 million hectares (ha) of swampland, accounting for 18% of Indonesia's land area (Table 1). The largest area is in Sumatra (13 million ha) followed by Kalimantan (10 million ha). A land evaluation conducted by the ministry of agriculture estimated that 7.5 million ha of swampland are available for cropping, with five million ha for paddy rice, 1.5 million ha for horticultural crops and about 1 million ha for perennial crops (Table 2) [20].

**Table 1.** Swampland area distribution in Indonesia (in million ha).

| Island | Type | | Total |
|---|---|---|---|
| | **Tidal Swamp** | **Non-Tidal Swamp** | |
| Sumatra | 3.02 | 9.91 | 12.93 |
| Jawa | 0.09 | - | 0.09 |
| Kalimantan | 2.99 | 7.04 | 10.02 |
| Sulawesi | 0.32 | 0.73 | 1.05 |
| Maluku | 0.07 | 0.09 | 0.16 |
| Papua | 2.43 | 7.44 | 9.87 |
| Indonesia | 8.92 | 25.20 | 34.12 |

**Table 2.** Swampland area availability for agriculture in Indonesia (in million ha).

| Island | Availability for (Mha) | | | Total (Mha) |
|---|---|---|---|---|
| | **Flooded/Paddy Rice** | **Horticultural Crops** | **Perennial Crops** | |
| Sumatera | 1.66 | 0.34 | 0.27 | 2.26 |
| Kalimantan | 0.85 | 0.53 | 0.48 | 1,85 |
| Sulawesi | 0.06 | - | - | 0.06 |
| Maluku | 0.08 | - | - | 0.08 |
| Papua | 2.47 | 0.60 | 0.19 | 3.26 |
| Indonesia | 5.12 | 1.47 | 0.93 | 7.52 |

## 4. Analysis of Swampland Development History in Indonesia

### 4.1. Indigenous People

Indigenous people living in swamp areas have utilized swamplands for agriculture for hundreds of years. In this paper, we focus on the Banjar people in Kalimantan, and the Bugis people who were originally from Sulawesi, but who settled in the coastal areas of Sumatra. During the wet season, there is plenty of fresh water available, and thus tidal swamps were used for rice cultivation [21].

Historically, indigenous people selected areas for cultivation based on topography, preferring areas that are higher and better-drained such as river levees and coastal ridges. They also used vegetation as indicators of suitable sites. Areas with dense foliage indicated favorable soil conditions. Nipa palm (*Nipa fruticans*) is an indicator of brackish or saline water, whereas sago palm (*Metroxylon* spp.) indicates fresh water. The Bugis people tended to avoid saline or brackish areas, while the Banjar people used both areas as long as the nipa growth was not too dense. The Banjar people avoided deep peats, but used areas surrounding peat domes as the domes provide good-quality water for irrigation [21].

After clearing trees and shrubs by slash and burn, the locals dug small ditches called handil. Handil, typically 2–3 m wide and 0.5–1.0 m deep, were constructed perpendicular to the river to protrude 2–3 km out from a stretch of a river. Handil can be extended up to 5 km allowing a land area of 20–60 hectares to be reclaimed. Water levels in rice fields were controlled with a gate (tabat) that was placed at the intersection of the primary and secondary channel. With this set-up, the Banjar community could maintain land productivity for up to 20 years [16].

The word handil comes from the Dutch word 'aandeel', meaning a share or part of the work. There are differences in the shape or pattern of handil made by the Banjar and the Bugis people. The Banjarese handil is generally straight as it is also used for boat transportation (local word: jukung). Bugis handil is generally winding to slow water loss when the tide is receding (Figure 4). Handil have several functions:

- Draining excess water and acidic water from agricultural land.
- Providing fresh irrigation water during tidal waves. This irrigation is limited to the topography and distance of the river to the land.
- Communication channel for transportation of people and goods.

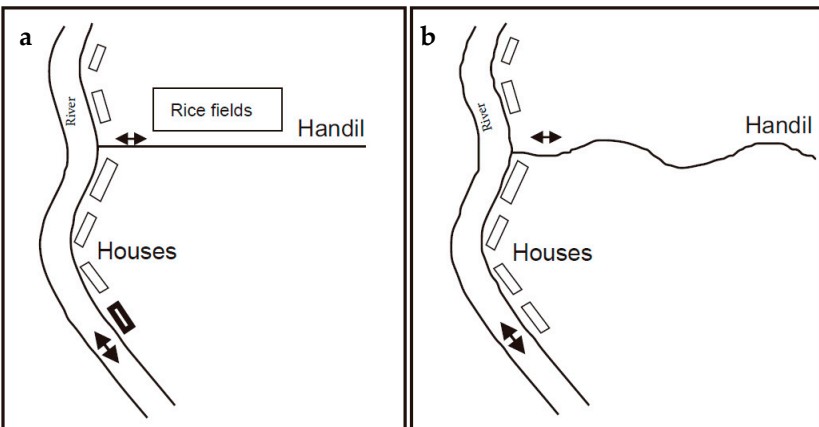

**Figure 4.** Handil shape or model of the Banjar people (**a**) and Bugis people (**b**), which can be found in Kalimantan and Sumatra swamps. Based on Reference [16].

Land preparation for rice cultivation involved clearing existing vegetation with a tajak, a sickle-like tool with a short handle. The weeds and bushes were left in the field for 10–15 days, then gathered into heaps, which were turned every week or so to aid decomposition. Once decomposed, the composts were spread over the land before planting [21]. The composts served as a source of nutrients, helped reduce evaporation, kept the underlying soil moist, and kept deeper soil—which have sulfidic materials—in a relatively reduced state. This traditional system is the equivalent of minimum tillage in Western agriculture [22].

The indigenous Dayak people in Kalimantan used peatlands which they called petak lawau for rice fields and had a rotating farming system. They divided the land into zones of settlement, bushes, paddy field (pahumaan), harvested paddy field (jurungan), plantations, sacred zones, and protected zones (kayuan) [16]. When the land was no longer fertile, the Dayak people practiced slash and burn, and moved to another place. After the land was abandoned for 1–7 years, it became bush and within 7–12 years the bush became a forest.

*4.2. Dutch Colonial Period (pre 1950)*

The Dutch colonial government began large-scale swampland development in South Kalimantan by digging large canals, known locally as anjir, to connect tidal rivers [23]. This development aimed to boost crop production after World War I. Anjir Serapat and Anjir Kelampang were hand dug around 1896. In 1935, the Dutch completed Anjir Serapat, which is 28.5 km long and connects the Kapuas Murung River with the Barito River. While anjir were built for transportation, the locals settled around the banks. Farmers made small handil every 200–500 m, extending up to 5 km in length, with the overall structure resembling a herringbone. From 1920–1962 around 65,000 ha of swamps were cultivated by farmers.

In 1941, the Dutch colonial government continued the development by building Anjir Tamban, which ran perpendicular to the Barito River. The area was part of a Dutch transmigration sponsored program with settlers from East Java. The work was halted because of world war and resumed in 1952, with a 25 km long canal extending to the Kapuas river. The farmers then started to build secondary canals to drain areas far from the anjir. Until 1940, there were 216 families with a total of 989 inhabitants in the area. These areas include Purwosari Village in the Tamban Subdistrict in Central Kalimantan, which is well-known as a center for coconut and rice production (Figure 5).

In the 1950s, H.J. Schophyus, a Dutch agricultural expert, and Haji Idak from Banjarmasin, continued the development of swamplands in Kalimantan [24]. Schophuys proposed the swampland be divided into separate water management units called polders. Schophuys and Idak designed the construction of polders to develop swamp in the Alabio, North Hulu Sungai Regency in South Kalimantan, a non-tidal zone along Negara river covering an area of 6500–7000 ha. This polder area is

now a center for rice production, however, only around 3000 ha can be planted each year. There were times when the areas could not be planted because of high standing water. Schophyus and Idak also developed tidal swamplands in the Mentaren area in Pulang Pisau Regency, Central Kalimantan, each area covering about 2300 ha.

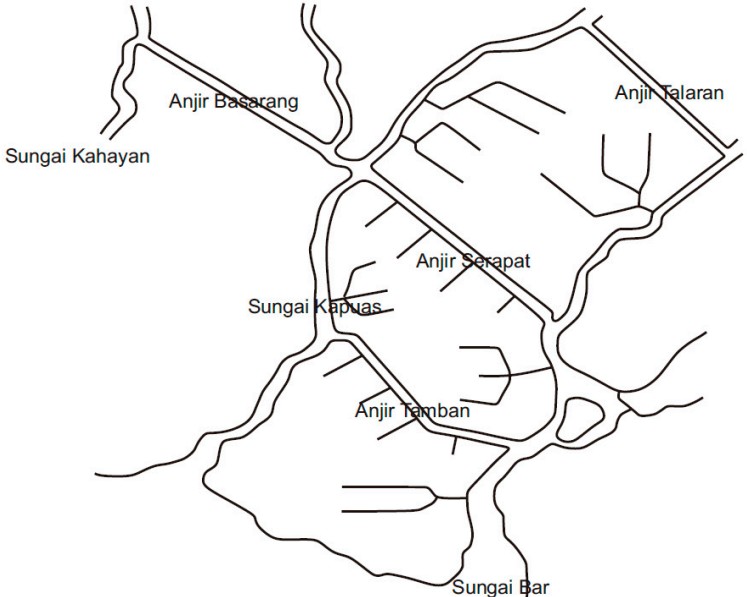

**Figure 5.** A sketch of the anjir system found in South Kalimantan and Central Kalimantan, based on Reference [18].

### 4.3. 1956–1958 Period

From 1956 to 1958, Indonesia faced a rice shortage. A rice intensification program in Java had not succeeded because of the limited irrigation system. Expanding upland was not viable, so focus turned to tidal swamp development. Ir. Pangeran Mohammad Noor, the Minister of Public Works and Labor of Indonesia (1956–1958) initiated swampland reclamation using the anjir system through the Dredge, Drain and Reclamation Project. This project made several large canals that connected two large rivers, and opened access to swamplands between the two rivers (Figure 4).

In this period, Anjir Serapat was further developed and a bridge was built to connect the Barito River (Banjarmasin city) and Kapuas Murung (Kuala Kapuas city). This anjir was also used by large boats and cargo ships as transportation between the two cities. The locals built secondary handil and the area became the center of rubber and rice production.

The anjir system was then replicated in several other areas in South Kalimantan, Central Kalimantan, West Kalimantan, and South Sumatra. Notably in Kalimantan, Anjir Talaran connects the Barito River in the Barito Kuala District with the Kapuas Murung River in Kapuas District (26 km). Anjir Kelampan connects the Kahayan River in the Pulang Pisau District with the Kapuas Murung River in the Kapuas District (20 km), and Anjir Basarang connects between the Kahayan River in the Pulang Pisau Regency and the Kapuas Murung River in the Kapuas District (24.5 km).

There was a plan to build an anjir in West Kalimantan to connect the Kuala Dua River with the Punggur Besra River (15 km) and the Jelai River with the Landak River (27 km). In South Sumatra, a plan was outlined to connect Sebokor and Cinta Manis rivers (21 km), between the Borang-Konton Rivers (5 km) and the Ogan-Kramasan Rivers (15 km). Due to the lack of infrastructure and funding these projects could not be realized. Nevertheless, in this era, transport via water became the key to the development of regions in Kalimantan. Transportation by sea, lake, river, and thousands of tributaries was the mainstay for the mobility of goods and residents because it was easy, cheap, and readily available. Residents and communities in coastal areas and riverbanks also became more connected.

The Ministry of Agriculture launched a three-Year Rice Production Plan to achieve food self-sufficiency in 1958. However, due to unstable political conditions, food self-sufficiency was not achieved.

*4.4. 1969–1995 Period*

This period consists of a series of five-year development programs called Pelita (Table 3). Starting in 1969, Pelita was established in response to urgent economic problems, including food sufficiency. In the first five-year development program (Pelita I, 1969–1974), the government began projects to reclaim large areas inland from coastal rivers. From 1970–1980, about 558,000 ha of tidal swamps were planted with rice, comprising 7% of the total rice area. This massive expansion of agriculture and resettlement of farmers via the transmigration program was funded by the world bank and aimed to reclaim more than 30,000 km of coastal areas in Sumatra. It was quoted as one of the world's major experiments with the use of marginal land [25].

**Table 3.** Swamp areas developed in Indonesia over 25 years as part of Indonesia's five-year-development plans (1969–1994).

| No. | Period of Development/Years | Areas Reclaimed (ha) | Transmigration (No. of Households) | |
| --- | --- | --- | --- | --- |
| | | | Government Sponsored | Voluntary |
| 1 | Pelita I (1969–1973) | 59,907 | 46,286 | - |
| 2 | Pelita II (1974–1978) | 268,997 | 84,639 | - |
| 3 | Pelita III (1979–1983) | 418,003 | 364,977 | 169,497 |
| 4 | Pelita IV (1984–1989) | 98,998 | 502,221 | - |
| 5 | Pelita V (1990–1994) | 54,088 | 180,000 | 370,000 |
| - | Total | 900,000 | 1,178,113 | 539,497 |

Source: [18].

Rice is highly suitable for swamps as rainfall is adequate and the land is close to populated areas meaning access to labor and markets [14]. Under the Directorate of the General of Water resources of the Department of Public Works, the Tidal Rice Development Project (P4S, Proyek Pengembangan Pesawahan Pasang Surut) was set up. This was followed by the P3S (Proyek Pengairan Pasang Surut) project. Two major drainage systems were developed:

● The fork or herringbone system (Figure 6) with a reservoir at the upstream end of the main canals, which can flush the canal.
● The comb system, a rectangular grid without a reservoir.

The project aimed to cover 5.25 million ha of swampland in Sumatra and Kalimantan in 15 years. However, by the end of the project, only 1.2 million ha were reclaimed. A network of 29 fork systems was created in South and Central Kalimantan and a network of 22 comb systems spread across Sumatra, and West Kalimantan, with a small portion in Sulawesi and Papua. This period was also marked by difficulties in developing inland swamps with potential acid sulphate and peat soils.

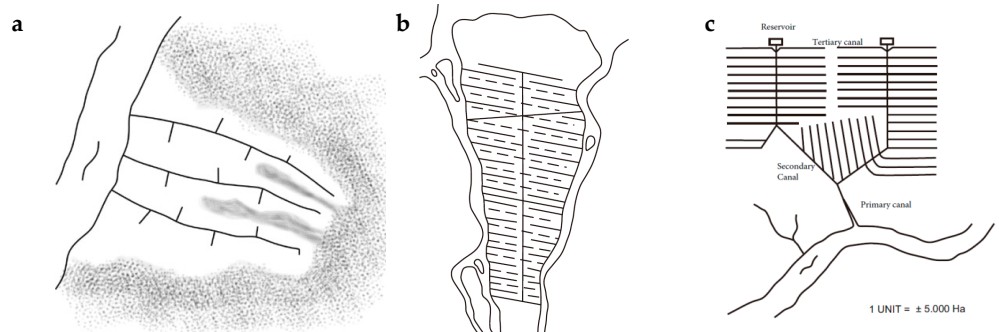

**Figure 6.** Canal design in tidal lands: (**a**) Indigenous system; (**b**) fork system; and (**c**) comb system. Based on Reference [23].

From the early 1970s to the mid-1990s, the government in Kalimantan and Sumatra began reclaiming tidal swamps. The primary goal was to increase rice production. Unfortunately, the reclamation of some swamp areas was not well managed. Poor logging and land clearing techniques, and excessive drainage via canals did not consider the danger of potential acid sulphate soils. Reclamation and settlement went together during the initial years when there was still a lack of knowledge on the area. Many transmigration developments on swamplands failed and were abandoned.

Other socioeconomic problems in the development of swampland were a shortage of labor for land clearing. It is not possible to use tractors on swamp soils. There were also problems with in-filling of existing anjir due to bank erosion. Some areas in Central and South Kalimantan had dead-end canals causing water stagnation, which increased acidity and toxicity.

From 1985–1990, the government focused on rehabilitating the canals and built water control structures. These canals and controllers were focused on drainage and many do not function according to agricultural needs. The government also intensified agricultural research and development activities, especially food crops on tidal swamps through various research and development projects known as Swamps II and ISDP Projects. Reclamation and development of swamps in this and the previous era did not produce good results due to a lack of soil knowledge, inadequate application of technology, and limited scientific support.

To overcome these problems, a staged development program was introduced. The first stage focused on rice production in smallholder farmers with an open tidal canal system in unripe and organic soils. The second stage involved maturing the soils, and social cohesion, water management, crop diversification, mechanization, and public and socio-economic services were introduced. The final stage employed a full water management system.

Reclamation in this era was supported by a transmigration program that placed residents from Java, Bali, and Nusa Tenggara in Kalimantan, Sumatra, Sulawesi, and Papua swamps. The swamp areas developed in this era were expected to use cultivation technology. Rice covers around 400,000 ha in Sumatra and 150,000 ha in Kalimantan. Some of these swamp areas have succeeded in becoming centers of rice, vegetable, and plantation production today.

Within 25 years (1969–1994), swamps that were planned and developed reached almost 1 million ha (Table 3). Of the 900,000 ha that were developed, around 715,000 ha are on tidal swampland and 185,000 ha on non-tidal swampland.

The expansion of swamp use for agriculture and transmigration succeeded in re-settling 1,717,610 households, just shy of the target of 2 million households. By 1995, around 1.18 million ha of swampland had been successfully reclaimed in seven major provinces. However, this program was criticized as the transfer of poverty from Java to Kalimantan and Sumatra [26].

*4.5. 1995–1999 Period*

The development of swamp areas in this period was due to the Presidential Instruction (Inpres) No. 6 of 1995, the One Million Hectares Peatland Development Project in Central Kalimantan. This project is better known as the Mega Rice Project in Central Kalimantan. The Presidential Instruction No. 6 of 1995 was issued on 5 June 1995, followed by a Presidential Decree No. 82 of 1995, Presidential Decree No. 83 of 1995, and Presidential Decree No. 84 of 1995.

This project was based on concerns over increasing land conversion following population growth, while rice production stagnated. Despite achieving food self-sufficiency in 1984, Indonesia again became an importer of rice from 1988–1989 [27].

The tidal swampland in Central Kalimantan is managed by an Inter-Department Team: Ministry of Public Works, Ministry of Agriculture, Ministry of Transmigration, Ministry of Forestry and National Land Agency.

The Mega Rice Project failed as it was hastily executed and mismanaged. Drainage networks were built without considering the land. One major error was excessively draining areas with peat domes.

As the domes act as a water reservoir, water supply diminishes. The excessive drainage also exposed some of the acid sulfate soils causing extreme acidification that made the soil toxic to plants [27]. Due to strong pressure from both within the country and internationally, Presidential Decree No. 80 of 1999 was issued and the Mega Rice Project was shut down. Large areas of swamplands that had been cleared were abandoned.

*4.6. 2000–2013 Period: Rehabilitation and Revitalization*

In this period, based on Presidential Instruction No. 2 of 2007, swamp development focused on rehabilitating areas of the ex-Mega Rice Project in Central Kalimantan Province, but there were no activities in the agricultural sector. Some areas still have potential for agriculture, and to provide incomes to 13,500 households (around 600,000 people) [28]. The uncertainty in efforts to revitalize the Mega Rice Project area has resulted in further land degradation due to land fires that occur every dry season, especially on uncultivated land.

*4.7. Lessons Learned*

Indonesia has learned a lot of lessons and gained much knowledge of swampland management through numerous reclamation projects, research, and development. The initial development of swamplands for transmigration settlement was done haphazardly with a low budget. This was further challenged by unknown soil conditions in the new areas. Unknown risks such as acid sulphate soils became expensive lessons in swamp management. The Javanese settlers in South Kalimantan, familiar with the fertile volcanic soil, had not encountered lowland with high-water levels. They faced great difficulties with these unfamiliar environments, which they described as 'water manipulates people'. Acquiring and developing knowledge about the new environment emerged as the most pressing task for their survival [29]. The failure of the Mega Rice Project was due to technical problems, as well as socio-economic and cultural issues [27].

Such lessons have similarity with on the wetland development in Vietnam [17], where they identified issues similar to Indonesia:

- The policies of settlement and economic development as primary indirect drivers.
- Agricultural growth and expansion, the availability of wetlands and their natural resources for exploitation are direct drivers.
- Ill-planned canal construction and infrastructure development, and others resulted in land degradation.
- The management of wetlands reflects the socioeconomic drivers at a particular time.
- Human modifications of wetlands intensify risks from droughts, floods and saline water intrusion.
- The inconsistent policies of different government regimes over time have complicated the management and conservation of the wetlands.

Thus, future development should consider a more holistic approach.

## 5. Staged Development of Swampland

From the history, and lessons learned, current and future development of swampland should follow a staged development with continued review and redevelopment to ensure a sustainable development. Lessons learned from Indonesia's history of swamp development can be grouped the into six categories: water management, land use and soil understanding, local participation, sectoral vs. integrative development, unified management, and indigenous knowledge.

Based on historical perspectives, we can now formulate technology that has worked in Indonesia. The technology is mainly in four areas: Water management, land use strategy, improved crop varieties, and crop management. Improved crop varieties and crop management are discussed in detail in Reference [30]. The following sections will only discuss water management, and land use strategy.

### 5.1. Water Management

Water management in swamplands should avoid over-drainage, which can result in the exposure of acid sulfate materials. In peatlands, over drainage can result in peat oxidation, subsidence, hydrophobicity that enhances fires, and carbon emissions. Water management should drain only the surface water to enable plant roots to survive. It is also important to leach out stagnant water, which accumulates acids and toxic elements [15].

The Indonesian Agency for Agricultural Research and Development has developed two water management systems based on operational use [31]. This includes the one flow direction water management system (Figure 7a), which supplies water from the primary canal to the field via a secondary and tertiary system. This system is mainly designed to leach out the acid accumulating in water. The second water management system, called the cascaded water table control system or tabat system (Figure 7b), is based on local knowledge. It is based on the traditional handil method that enables water to be retained during the dry season. The system has a gate that controls the water level.

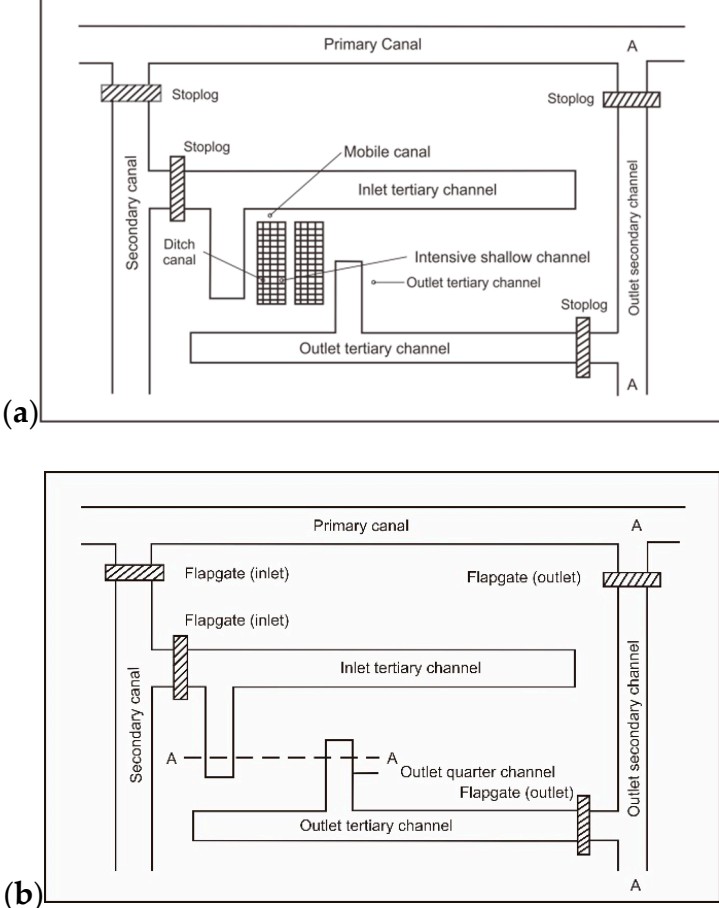

**Figure 7.** (**a**) One flow direction water management system for tidal land class A and B and (**b**) tabat water management system for class C and D (based on Reference [31]).

### 5.2. Land Use Strategy

Swamp areas are mostly potential acid sulphate soils. Reclamation that focuses on draining causes shallow groundwater during the dry season. Pyrite under inundation is stable, but upon exposure to the air can be easily oxidized resulting in severe acidification of soil and water. Furthermore, pyrite, Al, and Fe are toxic to plants. Haphazard drainage can also cause waterlogging in the rainy season [15].

There are two options to manage acid sulphate soils:

1. Leaching out pyrite oxidation products by rainfall or tidal flushing. This option is hard to achieve due to the required amount of fresh water, and it is a slow process.
2. Water management to minimize pyrite oxidation by keeping the groundwater table above the pyrite position. Irrigation is only required during the dry season [13].

Similarly, excessive drainage of peatland causes peat degradation. Surveying an area before development can avoid soil acidification and peat degradation. Maps of potential acid sulphate soils need to be produced via digital soil mapping techniques. The depth at which pyrite occurs also needs to be mapped out before conducting any drainage work [32]. Digital maps of acid sulphate soil distribution can support agriculture land sustainability and capability assessment. Areas with potential acid sulphate soils have are risky developments and need to be outlined in planning policies. Similarly, the map of peat thickness needs to be conducted with high efficiency and accuracy before development [33].

Using soil knowledge, we can design land use based on land type and crop requirements. The surjan system [19] provides a way of growing a range of crop in the same field via landscaping and land use design. The surjan system was originally from Java, consisting of alternately raised beds planted with dryland crops, and sunken beds planted with rice (Figure 8). The system is now used in South Kalimantan where farmers plant annual and perennial crops on the dry beds, while the sunken beds can be used for cultivating rice and fish.

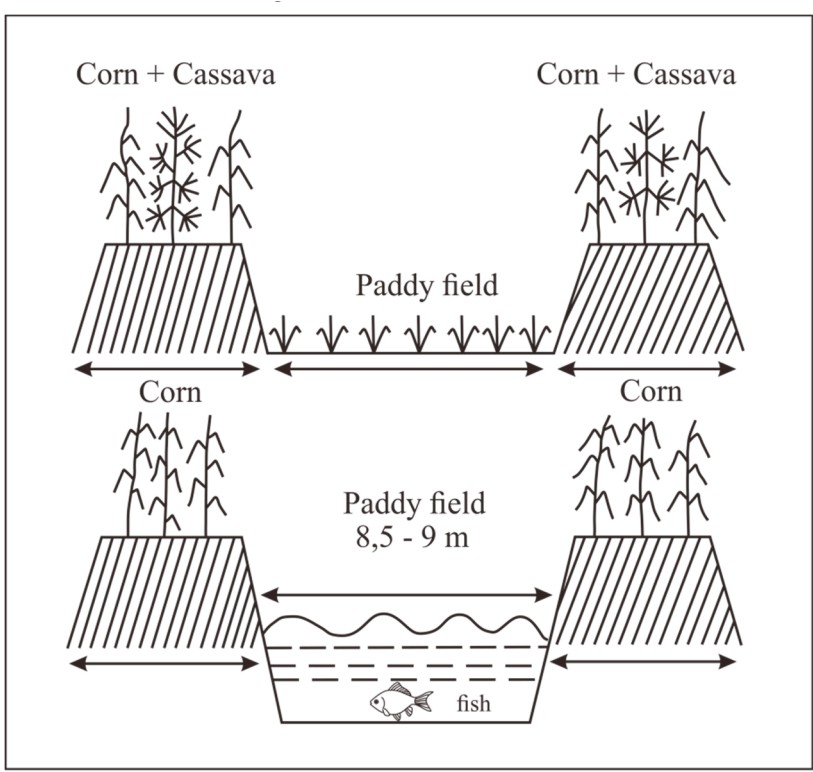

**Figure 8.** The surjan farming system. Top: Corn + Cassava and paddy field system, Bottom: Corn and paddy field intercropping (Source: Reference [34]).

In order to apply the technology, we need to consider local citizen participation, integrative development, unified management, and indigenous knowledge.

*5.3. Local Citizen Participation*

Residents who have lived in swampland for a long time have a balance between their lives and the nature around them. Their land management knowledge is based on an accumulation of several

generations of experience. They have developed a system of behavior that regulates the relationships between the people and their environment as a landscape [35]. Local communities have policies in managing the sustainability of channels and gates. For example, the subak system in the Balinese community, and local laws keep paddy fields productive and water remains available for decades [36].

The development of swamplands in the 1950s involved many residents in planning and implementation. In the 1990s, because of excessive trust in knowledge and technology, the important role of this local community was ignored. The development of future swamplands should involve local communities at each stage of development [37]. Society should be a subject, not an object, of development. The handil institution in the Banjar community proves how local culture can maintain and secure the sustainability of swamps [38].

### 5.4. Integrative Development

During the reclamation periods, swampland is targeted as a sectoral development, to produce rice. This single commodity-based or sectoral objective tends to fail. The Mega Rice Project is too commodity-oriented and are therefore not sustainable.

Integrative development aims to improve the welfare of people living in an area, compared to sectoral development, which only aims to obtain a specific output such as rice production. Commodity development only looks at crop productivity as opposed to the integrative development that focuses on land productivity. A study on a farming system in tidal land under the surjan system found that mixed citrus and coconut with rice farming resulted in the highest farmer income compared to rice monocropping [34].

In addition, integrative development considers that drainage channels should be made to not only regulate the water system, but also to provide new communication and transportation lines to open an area and bring agricultural products to the broader market. An integrated approach, including an integrated farming system, is key for success in swampland development [39].

In the future, the development of swamplands should be integrative from the start, including during planning. When more people are involved, more outcomes are obtained, and there is greater opportunity for sustainability from swampland development.

### 5.5. Unified Management

Management here means that the reclaimed swamplands should be managed for maximum output and increased welfare. Many ministries are involved in swampland management, and each has its own rules and interests. Often the importance of an institution is incompatible with the interests of other agencies.

Centralized management that coordinates the development should be established. All stakeholders should be working towards the same overarching goal. Past failures have largely hinged on weak coordination and communication between agencies and between communities. Local communities and young people need to be involved in the development and management of swamp areas.

### 5.6. Indigenous Knowledge

The indigenous Banjar, Bugis, Malay, Dayak, and Papua people have strong cultural attachments to swamps and water. They have lived in balance with their environment in the form of land, plants, and climate. They possess a work culture that is specific to its environment. They also have beliefs and rules about what is allowed and not allowed in the development of swamplands. This is a local decision that deserves respect and should be used to encourage the development of swamps according to their environment. Indigenous knowledge documentation is still rare and needs to be a focus in the future. The importance of indigenous knowledge for wetland development in Thailand is illustrated in Reference [40].

## 6. Conclusions

Indonesia has over 34 million ha of swampland which has a high potential to support Indonesia's food security program. Crop production and productivity can be increased by optimizing existing reclaimed and abandoned swamplands. Improved water management by constructing new and revitalizing existing infrastructures, especially drainage and water control systems, will further improve crops. Land productivity can also be improved by crop-fish-poultry diversification.

Lessons from over 60 years show the importance of involving local communities in the development and management of swamplands, and not to abandon local wisdom. Going forward, swampland development should focus on revitalizing existing drainage networks.

Water management is key to using the land sustainably, which includes keeping the potential acid sulphate soil inundated, removing excess surface water, and providing fresh water for leaching and diluting acidity. For reclaiming new areas and optimizing existing areas, precise planning is required including land-soil-water characterization and mapping, landscape and land use design, and an adaptive development approach. These combined efforts can increase crop production and productivity, as well as community welfare.

**Author Contributions:** A.A.S. and Y.S. conceived and contributed to the idea of the paper; Y.S. conducted the analysis; Y.S. and B.M. contributed to the discussion; A.A.S., Y.S. and B.M. wrote the manuscript.

**Funding:** This research received no external funding.

**Acknowledgments:** The authors acknowledge administrative and technical support from staff at Indonesian Center for Agricultural Land Resource Research and Development.

**Conflicts of Interest:** The authors declare no conflict of interest.

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
