# Peer review of "A Framework for the Development of Wetland for Agricultural Use in Indonesia"

_resources, doi:10.3390/resources8010034_

Round 1

Reviewer 1 Report

Review: A Framework for the Development of Wetland for Agriculture Use in Indonesia

 L12: with a shrinking arable land area, it is the only solution?

L33: add reference

L36: the second longest in the world, after …?

L55:60: rewrite the paragraph

L93: add reference

Figure2: please locate Indonesia in a world map, then locate the swamplands

Table 1 and 2: same title?

L361; please explain much ore about water management system

Please use more references through the paper.

Please add some examples out of Indonesia

Author Response

2nd February 2019

Dear Editor

We thank the editor and two reviewers for their valuable and constructive comments. We have revised our paper according to their comments and suggestions. We include below our responses to each of the comments. We also highlighted the changes in our manuscript via a “tracked changes” document.
We believe we have addressed all of the reviewers’ concerns and this paper is ready for publication.

Looking forward to hearing from you.

Sincerely yours

Dr. Yiyi Sulaeman, on behalf of the authors

Reviewer 2 Report

The manuscript by Sulaiman et al is an interesting and well written paper about how wetlands in Indonesia may be used in a sustainable manner for agriculture, specifically rice production. The paper provides important historical information while also presenting a holistic proposal for managing such systems. I have a few concerns, mostly dealing with language.

1.       The authors keep using the word “reclaimed” or a variant of that word in describing converting the wetlands for rice production. Reclaimed is usually used in terms of habitat restoration. I would urge the authors to use other verbiage such as converted instead of reclaimed.

2.       Line 40-41 is an incomplete sentence and is in need of a verb.

3.       There are numerous instances of redundant sentences or phrases. For example:

a.       Line 70-71. Omit the sentence “We describe swampland in Indonesia.”

b.      Line 92-93. Remove the end of the sentence “…although organic soils can often be found in swampland.”

c.       Lines 171-175: Much of the information in this paragraph was already stated and should be omitted.

4.       Other language problems:

a.       Line 193: is World War II (2)

b.      Line 234: “…river banks also became more connected.”

c.       Line 273: change shallowing to in-filling

d.      There are problems with the authors using the wrong verb tense starting with line 302 to the end of the text. Usually the authors use the present tense when the past tense is correct.

e.      Line 440: change to “…beliefs and rules about what is allowed and not allowed…”

f.        Line 437: Omit the words “people of” before Banjar…

5.       The paragraph at the end of section 4 (lines 339-342) should actually begin section 5, replacing or being combined with the original first paragraph of that section. I recommend that the authors re-write sections 5 and 6 by combining them into a more logical presentation. Most of the information already there is quite good and relevant but it can be presented in a much better manner.

Overall, this is a good paper which deserves to be published once the authors make the few recommended corrections.

Author Response

(The authors gave the same response as above.)
